# Applying a Radiation Therapy Volume Analysis Pipeline to Determine the Utility of Spectroscopic MRI-Guided Adaptive Radiation Therapy for Glioblastoma

Anuradha G. Trivedi [1,2], Su Hyun Kim [1], Karthik K. Ramesh [1,2], Alexander S. Giuffrida [1,2], Brent D. Weinberg [3,4], Eric A. Mellon [5], Lawrence R. Kleinberg [6], Peter B. Barker [7], Hui Han [8], Hui-Kuo G. Shu [1,4], Hyunsuk Shim [1,3,4,*] and Eduard Schreibmann [1,*]

1. Department of Radiation Oncology, Emory University School of Medicine, Atlanta, GA 30322, USA
2. Department of Biomedical Engineering, Emory University and Georgia Institute of Technology, Atlanta, GA 30332, USA
3. Department of Radiology and Imaging Sciences, Emory University School of Medicine, Atlanta, GA 30322, USA
4. Winship Cancer Institute, Emory University School of Medicine, Atlanta, GA 30322, USA
5. Department of Radiation Oncology, Sylvester Comprehensive Cancer Center, Miller School of Medicine, University of Miami, Miami, FL 45056, USA
6. Department of Radiation Oncology, Johns Hopkins University, Baltimore, MD 21218, USA
7. Department of Radiology and Radiological Science, Johns Hopkins University, Baltimore, MD 21218, USA
8. Biomedical Imaging Research Institute, Cedars-Sinai Medical Center, Los Angeles, CA 90048, USA
* Correspondence: hshim@emory.edu (H.S.); eschre2@emory.edu (E.S.); Tel.: +1-(404)-778-4564 (H.S.)

**Abstract:** Accurate radiation therapy (RT) targeting is crucial for glioblastoma treatment but may be challenging using clinical imaging alone due to the infiltrative nature of glioblastomas. Precise targeting by whole-brain spectroscopic MRI, which maps tumor metabolites including choline (Cho) and N-acetylaspartate (NAA), can quantify early treatment-induced molecular changes that other traditional modalities cannot measure. We developed a pipeline to determine how spectroscopic MRI changes during early RT are associated with patient outcomes to provide insight into the utility of adaptive RT planning. Data were obtained from a study (NCT03137888) where glioblastoma patients received high-dose RT guided by the pre-RT Cho/NAA twice normal (Cho/NAA $\geq$ 2x) volume, and received spectroscopic MRI scans pre- and mid-RT. Overlap statistics between pre- and mid-RT scans were used to quantify metabolic activity changes after two weeks of RT. Log-rank tests were used to quantify the relationship between imaging metrics and patient overall and progression-free survival (OS/PFS). Patients with lower Jaccard/Dice coefficients had longer PFS ($p = 0.045$ for both), and patients with lower Jaccard/Dice coefficients had higher OS trending towards significance ($p = 0.060$ for both). Cho/NAA $\geq$ 2x volumes changed significantly during early RT, putting healthy tissue at risk of irradiation, and warranting further study into using adaptive RT planning.

**Keywords:** spectroscopic MRI; survival biomarkers; glioblastoma; adaptive radiation therapy planning

## 1. Introduction

Glioblastoma is an aggressive brain cancer and is the most common primary brain malignancy [1,2]. Despite aggressive standard of care treatment for glioblastomas, the median survival is 15–16 months with frequent recurrence within 4–6 months [3]. One of the challenges that arises during glioblastoma treatment is precisely defining the radiation therapy (RT) target. Conventional imaging involves T1-weighted contrast-enhanced (T1w-CE) MRI and T2-weighted fluid-attenuated inversion recovery (FLAIR) MRI. T1w-CE identifies glioblastoma by a gadolinium-based contrast agent but fails to identify non-enhancing infiltrative tumor regions. FLAIR is not specific to tumors in that it identifies

abnormalities including edema, inflammation, and radiation effects [4–6]. Standard of care RT typically involves treating the resection cavity and the residual contrast enhancement on the T1w-CE MRI to a dose of 60 Gy and treating FLAIR hyperintensity to a dose of 45–54 Gy, followed by adjuvant temozolomide [3,7]. In a standard-of-care treatment course, the dosage and the treatment volumes are determined when the treatment is initiated, and are not altered during the course of treatment. However, for treating glioblastomas, the combination of T1w-CE and FLAIR MRIs to identify the RT target may not sufficiently identify infiltrative tumors and can thus result in undertreatment.

Combined with clinical imaging, magnetic resonance spectroscopy (MRS) measures metabolic concentrations that can identify infiltrative tumors without the use of contrast agents. An advanced form of MRS imaging, termed spectroscopic MRI, is a 3D whole-brain magnetic resonance spectroscopic imaging technique that identifies infiltrative, non-enhancing tumors with high specificity [8–11]. For glioblastoma, two metabolites of interest include choline (Cho) and N-acetylaspartate (NAA). Compared to healthy tissue, Cho levels are elevated in tumor tissue due to increased membrane synthesis in proliferating cells, and NAA levels are decreased due to loss of healthy neuronal function. Image-histology correlation studies have shown that the Cho/NAA ratio is a highly specific tumor biomarker [12].

Changes in metabolically abnormal regions throughout the course of radiation therapy may be useful to provide insight into response to treatment. To monitor volumetric changes, we have developed a pipeline that quantifies the change in treatment volumes during treatment and analyzes how this change is related to patient outcomes. In this report, we show the results of using our tool on a cohort of patients enrolled in a recently completed multisite clinical study (NCT03137888).

Information about volumetric changes in metabolic abnormality during treatment can provide insight into the potential utility of adaptive RT planning, a method not commonly used in glioblastoma treatment [13–17]. Adaptive RT planning is a strategy that modifies the radiation plan to accommodate changes in tumor and brain morphology during treatment to maximize the benefit of radiation fractions later in the therapy course [13]. The process involves modifying dose, field margin, field shape, and beam intensity according to variations in the treatment target during the early stages of RT. A key component of adaptive RT is the use of an imaging modality that can detect changes due to treatment early on. Since spectroscopic MRI directly images metabolic changes, it is ideally suited for this task as compared to conventional imaging. The application of our pipeline in this context will attempt to demonstrate the outcome prediction capability of spectroscopic MRI after two weeks of treatment (during week 3), determining the potential utility of using this modality to guide adaptive RT planning.

## 2. Materials and Methods

To perform an analysis of clinical and spectroscopic MRIs, we developed a quantitative tool that can provide volumetric and statistical comparisons of different RT volumes and metabolite maps. It is fully implemented in Python, primarily using pydicom and SimpleITK for reading DICOM files and performing image registration, lifelines for all statistical analysis, NumPy for all basic image operations, and matplotlib and The Visual Toolkit (VTK) for visualization [18–21].

### 2.1. Data Acquisition and RT Planning

The data used in this study were acquired as part of our multisite pilot study (NCT03137888) in which 30 newly diagnosed WHO grade IV glioblastoma patients were treated to an escalated dose of 75 Gy guided by metabolic abnormality identified on spectroscopic MRI. Details regarding enrollment and initial survival outcomes have been previously reported [22]. In this study, data from one patient was unavailable. Additionally, two patients had IDH mutations, which are no longer considered to be glioblastoma as of

2021 [23]. As a result, 27 patients were used for the analysis presented in this report. Of these 27 patients, 7 were MGMT promoter hypermethylated.

Approximately 3–4 weeks after resection and one week prior to starting RT (pre-RT), clinical (T1w-CE and FLAIR) and repeat spectroscopic MRIs were acquired. After each patient had completed two weeks of RT (mid-RT), spectroscopic, pre-contrast T1, and FLAIR MRIs were acquired. Post-contrast T1 MRIs were not acquired at mid-RT in order to avoid the cost and risk of additional contrast injection. All scans were processed and registered to T1 MRIs for each patient.

At each site, spectroscopic MRIs were acquired on a Siemens 3T scanner with GRAPPA-accelerated echo-planar spectroscopic imaging. A 20- or 32-channel head and neck coil was used, and the data were acquired with 50 ms echo time (TE), 1551 ms repetition time (TR), and 71° flip angle [22].

Pre- and mid-RT Cho/NAA abnormal volumes were created by contouring regions greater than or equal to two times the average Cho/NAA (Cho/NAA $\geq$ 2x) value in normal-appearing white matter that is contralateral to the tumor. Volumes were generated automatically in the Brain Imaging Collaboration Suite (BrICS) from each patient's spectroscopic MRI scans acquired one week prior to starting RT and two weeks after starting RT, respectively [24]. Both contours were manually checked and edited by MRS experts based on spectral quality as previously reported [24]. The pre-RT anatomic image used for registration was the T1w-CE and the mid-RT anatomic image used was the pre-contrast T1 since T1w-CE was not collected mid-RT. To perform the analysis depicted in Figure 1, the Cho/NAA $\geq$ 2x contours and the corresponding T1s and metabolite maps are inputted to the pipeline after being exported from BrICS as a series of DICOM files.

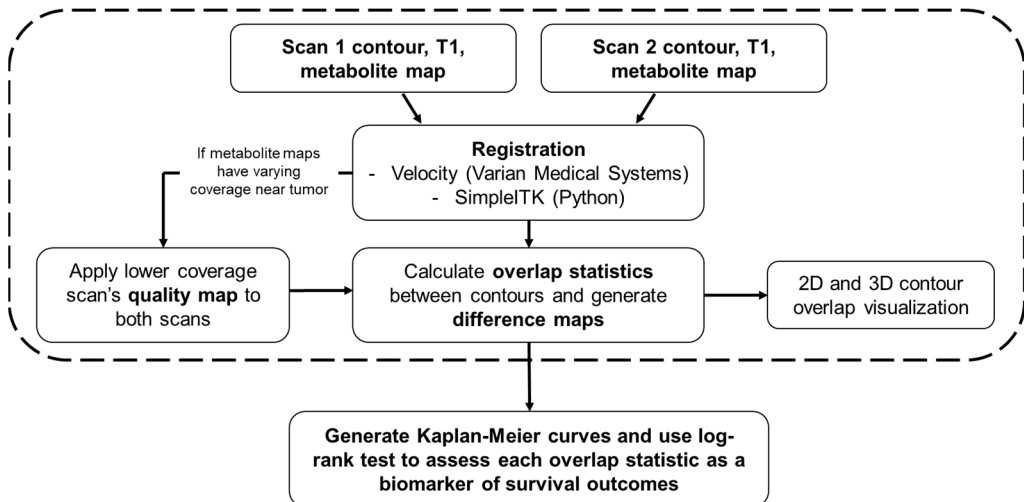

**Figure 1.** A flowchart detailing every component of the pipeline is described in this report. The user first inputs a pair of contours, T1s, and metabolite maps. All images from the second scan are registered to the first scan either in Velocity (Varian Medical Systems) or within the program using SimpleITK. After approval of registration, if metabolite maps have varying coverage in the vicinity around the tumor, then the quality map for the lower coverage scan is applied to both scans to maintain coverage across both metabolite maps. Overlap statistics between the two contours are then calculated, and difference maps are generated for each axial slice based on the new, modified metabolite maps. After this step, 2D and 3D renderings of the contours can be generated. The steps within the dashed box must be repeated for each pair of contours within a cohort of patients. The final step is to generate Kaplan–Meier curves for OS and PFS for each overlap statistic, stratifying the groups by the median value of the statistic of interest. A log-rank test is performed on each pair of Kaplan-Meier curves to determine if the overlap statistic used is a statistically significant biomarker of survival.

The pre-RT volume combined with the residual contrast enhancement from the T1w-CE received an escalated dose of 75 Gy, while the FLAIR abnormal regions received 50.1 Gy and the resection cavity received 60 Gy. More details regarding patient treatment have been previously reported [22]. Upon completing RT, follow-up imaging acquired every 2–3 months was used to determine PFS [25]. OS was determined based on communication with the patients and their oncology team as well as chart review.

### 2.2. Registration

Rigid and/or deformable registration was performed either using the Velocity software (Varian Medical Systems) or our SimpleITK registration implemented in Python. Both Velocity and our implementation have options to perform linear registration and deformable registration. Registration can be checked visually with a blend of the two images. Depending on the time between scans and the degree of morphology change, linear registration may be sufficient. In order to proceed with the automatic analysis, every subject's registration must be approved by the user, after which the second scan is re-sampled with the registration transform applied and saved.

The T1 from the mid-RT scan was first registered to the T1w-CE from the pre-RT scan. Once approved, the transformation was applied to all remaining data (contours and metabolite maps) associated with the mid-RT scan. All results were carefully inspected by two authors (A.G.T. and S.K.) and approved by the board-certified medical physicist (E.S.) who is specialized in image registration. The registered images were resampled and then stored in addition to the original images and were used for all remaining analyses.

### 2.3. Metabolite Map Coverage

For spectroscopic MRI maps, during initial data processing, poor-quality spectra were removed based on various criteria [26–31]. Voxels with a metabolite linewidth greater than 18 Hz were removed due to poor quality as an initial filtering step. For data acquired during this trial, the brain coverage was 70–75% after automatic artifact filtration to remove poor-quality spectra.

Such filtering may result in differences in coverage between the scans in areas surrounding the tumor. This process generates a quality map, which is a mask identifying the voxels that passed all criteria. To allow for unbiased comparative analysis, the quality maps with lower coverage masks were applied to the metabolite maps and any contours generated based on metabolic abnormality for both scans.

### 2.4. Overlap Statistics

To quantify volumetric differences in contours, we calculate a variety of spatial overlap statistics [32]. For binary mask contours $A$ (earlier scan) and $B$ (later scan), total overlap is calculated by identifying the number of pixels in the intersection of $A$ and $B$ divided by the number of pixels in $B$. Jaccard coefficient (union overlap) is two times the number of pixels in the intersection of $A$ and $B$ divided by the number of pixels in the union of $A$ and $B$. Dice coefficient (mean overlap) is two times the number of pixels in the intersection of $A$ and $B$ divided by the sum of the total pixels in $A$ and the total pixels in $B$. The final metric calculated is Hausdorff distance, which represents the maximum distance of $A$ to the nearest point in $B$ [33]. The final quantities calculated are the direction and percentage of volume change. Neither quantity takes any spatial information into account, they only consider the change in volume. Volume change is calculated by taking the absolute value of the difference between the volume of $A$ and $B$ divided by the volume of $A$. Direction information can be preserved by not taking the absolute value of the volume differences.

### 2.5. Visualization

To visually validate the results of the overlap statistics, 2D and 3D visualizations of the inputted contours and metabolite maps are generated. 2D visualization involves overlaying the contour on top of the corresponding T1 and metabolite map, and 3D visualization

renders the two contours on the same plane so that differences in the contours can be identified. Additionally, the difference between normalized maps can be visualized, and are saved as 3D NumPy arrays. An example of 2D and 3D visualization, including a difference map, is shown in Figure 2. When running the 3D visualization code, contours can be rotated and zoomed in/out as well as altered in their opacity. A demonstration of the interactive 3D visualization is available in the online Supplemental Video S1.

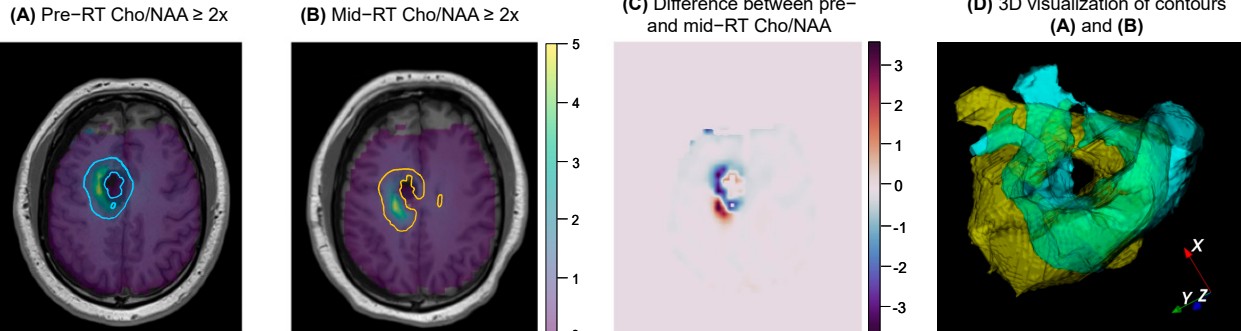

**Figure 2.** An example of the 2D and 3D visualization generated by the pipeline for a given subject. (**A**,**B**) show the pair of contours, T1s, and metabolite maps that were inputted by the user. (**C**) shows the difference between the two Cho/NAA maps for the same axial slice shown in (**A**,**B**). (**D**) shows a 3D overlay of the two contours (pre−RT shown in yellow, mid−RT shown in blue), which can be interacted with when the 3D visualization code is run.

*2.6. Statistical Analysis*

To determine if any of the previously described overlap statistics can be used as biomarkers of survival, we analyzed each one using Kaplan-Meier curves for OS and PFS [34]. For each biomarker of interest, the patient population is first stratified by the median biomarker value, resulting in two equal-sized subgroups. Kaplan-Meier curves for OS and PFS are generated for each subgroup, and a log-rank test is used to test for the difference between the survival distributions of the two subgroups [35]. A significant *p*-value indicates the overlap statistic being used can be considered a significant biomarker of survival [34].

Excluding information about spatial volume change, OS and PFS Kaplan-Meier curves are also generated for subjects with Cho/NAA $\geq$ 2x volume growth and reduction. The percentage of volume change can be tested as an additional biomarker with OS and PFS Kaplan-Meier curves stratified by median volume change.

*2.7. Combined Biomarkers*

The combination of two biomarkers may result in the identification of a more robust biomarker of survival. Each spatial overlap statistic can be combined (multiplied) with the direction of volume change (increase or decrease in volume) and the actual percentage change in volume. These combinations of biomarkers can also be evaluated with Kaplan-Meier curves and log-rank tests to be identified as possible biomarkers of survival.

**3. Results**

Between the pre- and mid-RT Cho/NAA $\geq$ 2x volumes of all 27 patients, the median total overlap was 0.796 (range: 0–0.964), the median Jaccard coefficient was 0.224 (0–0.723), median Dice coefficient was 0.365 (0–0.839), median volume similarity was −0.142 (−1.54–1.77), and median Hausdorff distance was 20.8 mm (8.54–92.2 mm).

Figure 3A,B show Kaplan–Meier curves of OS and PFS with the Dice coefficient as a biomarker. In Figure 3A, 27 patients were stratified by those with a Dice coefficient below and above the median (0.37), and Kaplan–Meier curves of OS were generated for each subgroup. The curves shown in 3A have minimal cross-over and large separation between

them, indicating that the Dice coefficient may be a good biomarker of OS. This is further supported by the *p*-value trending towards significance (*p* = 0.060) calculated by a log-rank test. Similarly, in Figure 3B, 27 patients were stratified by those with a Dice coefficient below and above the median (0.37), and Kaplan–Meier curves of PFS were generated for each subgroup. With no cross-over and large separation between the curves, it was found that patients with lower Dice coefficient had a higher PFS that was of statistical significance (*p* = 0.045). The same result and *p*-value were observed for the Kaplan-Meier plots for patients stratified by median Jaccard coefficient, shown in Figure 3C,D since stratifying by median resulted in the same subgroups for both Dice and Jaccard coefficients.

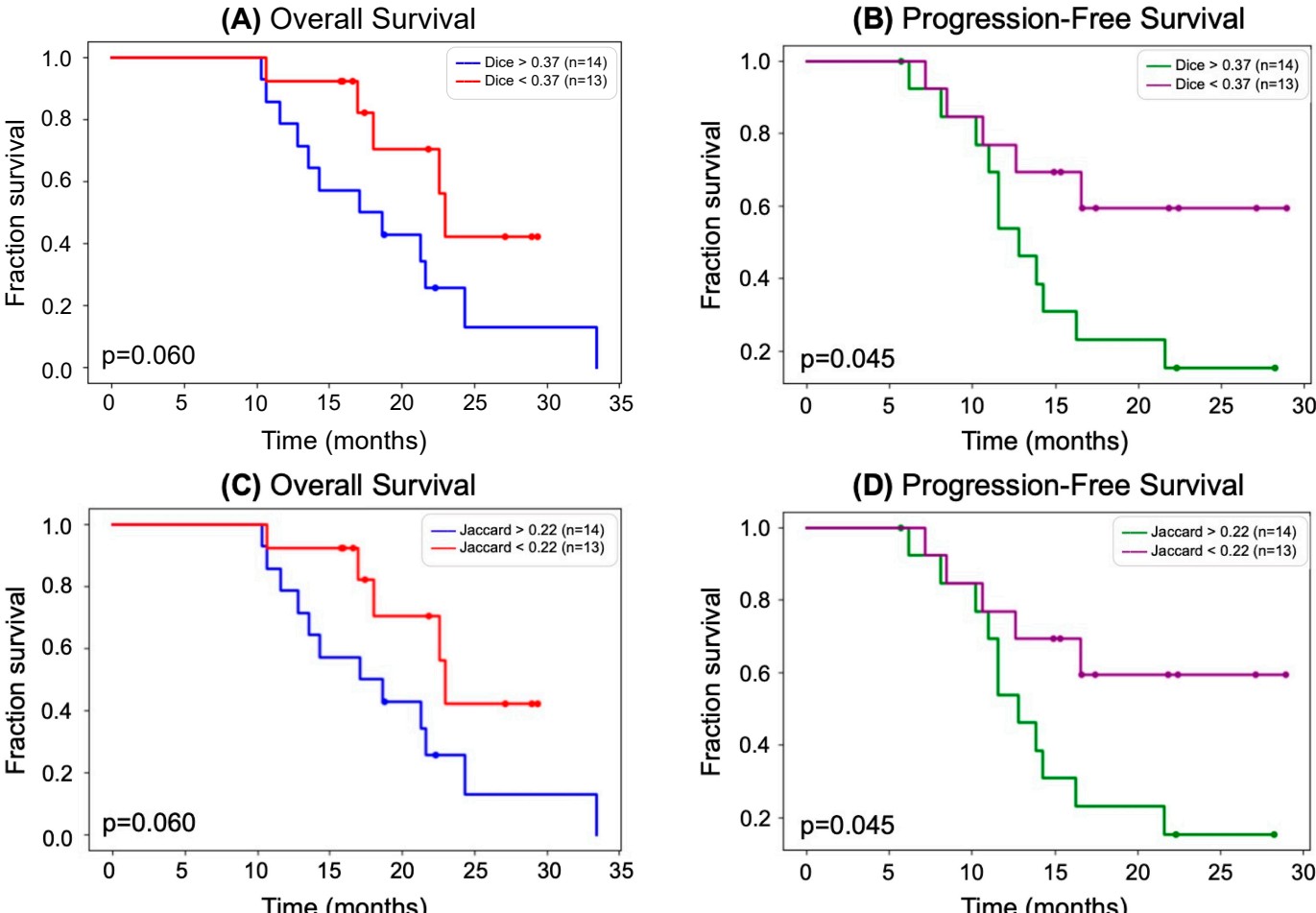

**Figure 3.** (**A**) Kaplan–Meier curves showing OS stratified by median Dice coefficient. Patients with a higher Dice coefficient had a lower OS. The separation between the curves was determined to be not significant by a log-rank test; however, the results trend towards significance (*p* = 0.060) and may be proven to be significant by a larger cohort. (**B**) Kaplan–Meier curves showing PFS stratified by median Dice coefficient. Patients with a higher Dice coefficient had a lower PFS. The separation between the curves was determined to be significant by a log-rank test (*p* = 0.045). (**C**) Same as (**A**) but instead stratifying OS Kaplan–Meier curves by median Jaccard coefficient. (**D**) Same as (**B**) but instead stratifying OS Kaplan–Meier curves by median Jaccard coefficient.

We also assessed combinations of biomarkers with OS and PFS Kaplan–Meier curves. Total overlap, Dice coefficient, and Jaccard coefficient were each combined with (1) direction of volume change and (2) percentage of volume change. Out of the 27 patients, 13 experienced an increase in Cho/NAA $\geq$ 2x volume, and 14 experienced a decrease in Cho/NAA $\geq$ 2x volume after completing two weeks of RT. When analyzed as potential

biomarkers of survival, incorporating directionality resulted in no statistically significant difference between groups.

Statistical analysis between pre- and mid-RT Cho/NAA ≥ 2x volumes showed notable volumetric/spatial changes in several cases. One such case is shown in Figure 2. Although the difference in volumes was very small (0.85 cc, −2% change indicating a slight decrease in volume after two weeks of RT), the total overlap between the two contours was only 57.7%. The difference map in Figure 2C depicts regions of positive change in blue (improvement in Cho/NAA values) and negative change in red (new or worsening areas of Cho/NAA abnormality). Another case is depicted in Figure 4, where the pre-RT scan Cho/NAA ≥ 2x contour volume was 1.00 cc, and after two weeks of treatment, this volume had grown to 4.96 ccs. These volumes had a 51.6% overlap, Jaccard coefficient of 0.063, Dice coefficient of 0.118, volume similarity of −1.54, and Hausdorff distance of 13.7 mm.

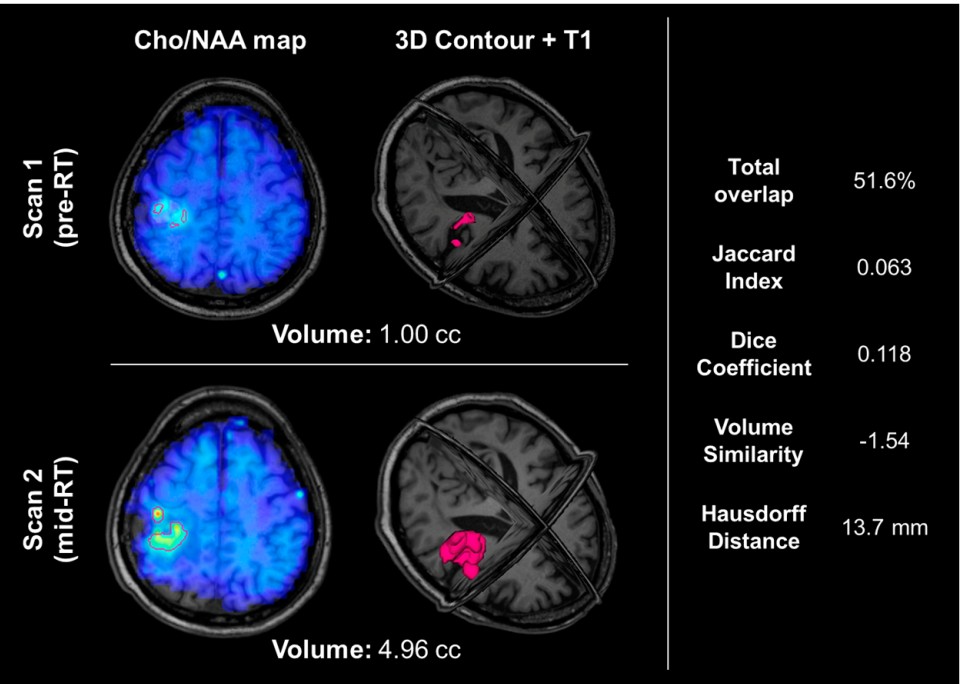

**Figure 4.** An example of a patient whose Cho/NAA ≥ 2x contour volume increased by almost 400%. The top row shows their pre-RT Cho/NAA ≥ 2x contour and Cho/NAA map overlaid on the T1, as well as a 3D visualization of the Cho/NAA ≥ 2x contour. The bottom row shows their mid-RT contour, Cho/NAA map, and T1. The right-hand side shows the overlap statistics calculated for the two contours displayed. This patient did not have an IDH mutation and was not MGMT promoter hypermethylated, and experienced progression 8.3 months after resection.

## 4. Discussion

In this paper, we used data from a recently completed study that used spectroscopic MRI to guide dose-escalated RT to assess the degree of treatment-induced metabolic changes in the early stages of therapy and investigate how these changes are related to survival outcomes. We used spatial overlap statistics including total overlap, Jaccard coefficient, Dice coefficient, volume similarity, and Hausdorff distance to quantify how metabolic activity changes after two weeks of treatment. We then used these metrics to identify a relationship between the degree of change in abnormal metabolic activity and survival in the context of motivating the use of adaptive RT planning.

Out of the five different spatial overlap statistics we calculated and tested as biomarkers of survival, the only statistically significant relationships were between the Dice coefficient and PFS ($p = 0.045$) and between the Jaccard coefficient and PFS ($p = 0.045$). This $p$-value is likely attributed to the early separation of curves (around 6 months), no cross-

over, and large separation beginning around 11 months. As shown in Figure 3B,C, a Dice coefficient of more than 0.37 or a Jaccard coefficient of more than 0.22 was associated with a shorter PFS (median of 18.0 months). This suggests that identifying tumors that changed more serves as a surrogate for early response to radiation therapy. While the relationships between Dice and Jaccard coefficients and OS ($p = 0.060$ for both) were not statistically significant, they trended towards significance. Although there was a separation between the OS curves for the Dice coefficient (Figure 3A) and Jaccard coefficient (Figure 3C), the slight cross-over of curves likely resulted in an insignificant $p$-value. We believe that a larger cohort is necessary to further investigate this relationship. Kaplan–Meier curves of the patient cohort stratified by all other overlap measures resulted in larger, insignificant $p$-values. Additionally, combining the direction of volume change (increase or decrease of volume) as well as the degree of volume change (percentage) with total overlap, Dice coefficient, and Jaccard coefficient did not result in the identification of statistically significant biomarkers, likely due to the small sample size.

The patient shown in Figure 4 experienced a nearly 400% increase in metabolic abnormality volume, indicating that this patient's tumor was likely resistant to RT. This is further supported by a relatively short PFS of 8.3 months and a lack of positive prognostic molecular data; this patient did not have an IDH mutation or MGMT promoter hypermethylation [8,36].

Patients with a decrease in Cho/NAA $\geq$ 2x volume could potentially have the size of their RT treatment field decreased in later treatment. Patients with an increase in Cho/NAA $\geq$ 2 volume could have their treatment altered, either with a larger/different treatment field or with experimental therapies such as new chemotherapy. While this result suggests evidence of a newly proliferating tumor that goes untreated with the original RT plan, it also emphasizes the potential value of a spatial volume overlap analysis tool for providing more insight into volumetric comparisons and showing changes due to treatment both quantitatively and visually. This pipeline can be used with other MRI modalities and volumes such as T1w-CE, FLAIR, and DWI. It can also be used with longitudinal data, providing statistical volume comparisons for more than two studies.

Limitations of this study include a small cohort and varying spectroscopic MRI quality. Since the same quality map had to be applied to both scans, by definition, some voxels will be lost, affecting the spatial overlap statistics and all statistical analyses. We are currently in the process of developing new shim-RF head coils that aim to overcome the issue of B0 field inhomogeneity in order to provide improved brain coverage and spectral quality [37]. With only 27 patients, we saw certain trends toward significance, and we believe that a larger study would enable us to investigate these trends more.

## 5. Conclusions

The analysis technique developed and demonstrated in this report can identify volumetric changes in the volume of metabolic abnormality during the early stages of RT and relate these changes to survival outcomes. This pipeline can be applied to a variety of modalities; in this report, we demonstrated this technique with spectroscopic MRI data collected for 27 patients pre-RT and after two weeks of RT to assess the utility of adaptive RT planning. Adaptive RT planning could potentially limit irradiation of healthy tissue over the course of RT, however; for this dataset, we determined that larger volumetric change indicated response to RT rather than evidence that adaptive RT planning could be useful.

**Supplementary Materials:** The following supporting information can be downloaded at: https://www.mdpi.com/article/10.3390/tomography9030086/s1, Video S1: Demo of 3D contour visualization.

**Author Contributions:** Conceptualization, E.A.M., L.R.K., H.-K.G.S. and H.S.; formal analysis, A.G.T. and S.H.K.; funding acquisition, H.H. and H.S.; methodology, H.S. and E.S.; software, A.G.T., S.H.K., K.K.R. and E.S.; supervision, B.D.W., H.S. and E.S.; validation, B.D.W., E.A.M., P.B.B., H.S. and E.S.; visualization, A.G.T., S.H.K., A.S.G. and E.S.; writing—original draft, A.G.T., S.H.K., H.S. and E.S.; Writing—review and editing, A.G.T., S.H.K., K.K.R., A.S.G., B.D.W., E.A.M., L.R.K., P.B.B., H.H., H.-K.G.S., H.S. and E.S. All authors have read and agreed to the published version of the manuscript.

**Funding:** This research was funded by NIH U01 CA264039 (E.A.M. and H.S.), NIH F31 CA247564 (K.K.R.), and NIH R01 NS121544 (H.H.).

**Institutional Review Board Statement:** The study was conducted according to the guidelines of the Declaration of Helsinki and approved by the Institutional Review Board (or Ethics Committee) of Emory University (IRB00094188 on 14 April 2017).

**Informed Consent Statement:** Informed consent was obtained from all subjects involved in the study.

**Data Availability Statement:** No new data were created or analyzed in this study. Data sharing is not applicable to this article.

**Conflicts of Interest:** The authors declare no conflict of interest.

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
