# Peer review of "Applying a Radiation Therapy Volume Analysis Pipeline to Determine the Utility of Spectroscopic MRI-Guided Adaptive Radiation Therapy for Glioblastoma"

_tomography, doi:10.3390/tomography9030086_

Round 1

Reviewer 1 Report

The manuscript entitled 'Applying a radiation therapy volume analysis pipeline to determine the utility of spectroscopic MRI-guided adaptive radiation therapy for glioblastoma'

presents results obtained in a multi-center study, which had not been included in a previous publication A multi-institutional pilot clinical trial of spectroscopic MRI-guided radiation dose escalation for newly diagnosed glioblastoma - PMC (nih.gov)

The overall goal of the multi-center study was to deliver an additional RT boost to a planning target volume (PTV) based on results obtained from an MRSI measurement, besides the PTV targets derived from contrast-enhanced T1 weighted MRI and FLAIR.  In the previous study, the overall survival time was longer than that usually observed in glioblastoma patients, indicating that the MRSI guided RT boost can improve PFS.

In the present manuscript, additional evaluations are presented in a quest to find a method that could be used to improve an on-going treatment, based on personalized decisions regarding changes in the PTV for the RT-boost. The authors performed an additional MRSI measurements at mid-time during the RT treatment. It appeared that the spatial location of the MRSI region was not stable during the treatment. This is a very interesting finding. Moreover, they found that the more the area had changed, the shorter the OS. These observations point towards the possibility, that the originally planned PTV is sub-optimal in these patients, and future studies may show that an update of the PTV to better match the volume identified at mid-RT can be beneficial.

Although interesting, the findings based on MRSI should be complemented with reports regarding changes in the volume detected by the more conventional contrast-enhanced T1w MRI and FLAIR methods. In this way  a more complete picture could be gained and  a putative link between the MRSI findings and the other MRI outcomes could be investigated. This would allow to obtain additional information and may better highlight a putative added value of the MRSI measurement.

Most patients did not have any IDH mutation, which is advantageous for the OS, but it would still be interesting to know whether also these patients benefit from the RT boost, or if it is superfluous, and more recommendable to use immunotherapy for this patient group. A discussion of this topic would be of interest.

In conclusion, this is an interesting study, and well worth to be strengthened by including additional parameters from conventional MRI measured.

Reviewer 2 Report

The authors present a pipeline for contouring and registering volumes of abnormality on spectroscopic imaging maps before and after radiotherapy. In theory this could perhaps be used for adaptive planning of RT dose volumes, but the application here was to assess whether there had been changes in response to RT, and to correlate such changes with outcome. They found that a change in volume of spectroscopic abnormality was predictive of longer survival.

 1. It is surprising that spatial shifts were found to enhance survival regardless of whether the overall volume of abnormality increased or decreased. Indeed, only 1 of the 4 metrics of overlap investigated did show significance, so it might have been a chance finding. Was there any attempt to look at combining the direction of volume change with the spatial shift metrics? It seems likely this might give a more robust marker. For example the subject in Fig 4 might no longer be an outlier where low overlap is linked with poor survival if the information were incorporated that the volume of abnormality was growing not shrinking.

2. It is however unclear whether the different median survival reported for volume increase vs decrease is significant. What do these Kaplan-Meier curves look like? What is the p-value?

3. Was the 2% change in volume reported in Fig 2 an increase or decrease?

4. The examples shown are all in regions of the brain easy to shim. Is this technique applicable in tumours located in the temporal and anterior frontal lobes?

Reviewer 3 Report

The authors analyzed imagistic data obtained from a phase I clinical trial where newly-diagnosed glioblastoma patients received high-dose conformal RT guided by pre-RT spectroscopic MRI (sMRI) metabolite maps based on Cho/NAA ≥2x volumes. Accordingly, all patients in this pilot study were treated to an escalated dose of 75 Gy which was guided by the metabolic abnormalities identified by sMRI readings. The patients received concurrent chemoradiation and sMRI scans pre- and mid-RT (i.e., before and after 2 weeks of chemoradiation). Spatial overlap statistics between pre- and mid-RT scans were used to quantify tumor metabolic activity changes after two weeks of RT. Key metrics were calculated pertaining volumetric differences in the contours of metabolic maps such as the total overlap and Jaccard/Dice indices. Log-rank tests were used to quantify the relationship between imagistic data and patient overall and progression-free survivals (OS/PFS). Upon completion of RT, follow-up imaging acquired every 2-3 months was further used to determine PFS. Key findings are: (1) patients with less contour overlap of metabolic activity over two weeks of treatment had longer OS, and (2) patients with lower Jaccard/Dice indices had higher PFS trending towards significance. The study suggests that changes in metabolically abnormal regions over the short course of two weeks of RT may be useful in providing insight into therapy responses, with larger volumetric changes in metabolic abnormality correlating with response to RT. While I find the study very interesting, I have a few comments/questions for the authors.  

1.     The authors found that the analyzed Cho/NAA ≥2x volumes changed significantly during these first two weeks of chemoradiation. I am wondering about the contribution of the inflammatory infiltrate to the Cho/NAA signal. It is well established that radiotherapy can trigger both pro-tumor (myeloid) inflammatory infiltrate as well as anti-tumor lymphocytic infiltrate (including CTL effectors). A discussion is therefore warranted in this regard because it is possible for some of those larger volumetric changes in metabolic abnormality that correlate with response to RT to be driven by adscopal (and even abscopal) anti-tumor immune effects.    

2.     The authors conclude that “for this dataset we determined that larger volumetric change indicated response to RT rather than evidence that adaptive RT planning could be useful” (lines 292-293).However, it is not entirely clear how the authors plan to further investigate the utility of their method for adaptive RT planning in the future or the potential caveats of their method as a tool for adaptive RT planning. Perhaps the authors could add a small paragraph detailing these future plans. 

3.  The Kaplan-Meier data showing PFS stratified by Dice coefficient indicate that the separation of the curves did not reach statistical significance, which was attributed to a lack of separation during the first 10 months. However, the relative contribution of tumor MGMT status to these curves is not clear. MGMT is one known mechanism of resistance to chemoradiation (i.e., TMZ cannot radiosensitize MGMT proficient tumors). Although the patient sample size for this study is relatively low (n=29), an attempt to further breakdown the patient PFS data based on the methylation status of the tumor MGMT promoter might be meaningful and very informative for the potential reader. Depending on the patient sample size, the data from those patients with methylated MGMT promoters might already show a statistically significant separation of PFS curves stratified by Dice coefficient during the initial 10 months.   

Round 2

Reviewer 2 Report

The manuscript is improved by the revisions.